# Associations of Fecal Microbiota with Ectopic Fat in African Caribbean Men

**DOI:** 10.3390/microorganisms12040812

**Published:** 2024-04-17

**Authors:** Curtis Tilves, Noel T. Mueller, Joseph M. Zmuda, Allison L. Kuipers, Barbara Methé, Kelvin Li, John Jeffrey Carr, James G. Terry, Victor Wheeler, Sangeeta Nair, Iva Miljkovic

**Affiliations:** 1Department of Epidemiology, Colorado School of Public Health, Aurora, CO 80045, USA; noel.mueller@cuanschutz.edu; 2LEAD Center, Colorado School of Public Health, Aurora, CO 80045, USA; 3Department of Pediatrics, Colorado School of Medicine, Aurora, CO 80045, USA; 4Department of Epidemiology, University of Pittsburgh, Pittsburgh, PA 15261, USA; zmudaj@edc.pitt.edu (J.M.Z.); kuipersa@edc.pitt.edu (A.L.K.); miljkovici@edc.pitt.edu (I.M.); 5Center for Medicine and the Microbiome, University of Pittsburgh, Pittsburgh, PA 15213, USA; metheba@upmc.edu (B.M.); lik2@upmc.edu (K.L.); 6Department of Radiology and Radiological Sciences, Vanderbilt University Medical Center, Nashville, TN 37232, USA; j.jeffrey.carr@vumc.org (J.J.C.); james.g.terry@vumc.org (J.G.T.); sangeeta.nair@vumc.org (S.N.); 7Tobago Health Studies Office, TTMF Jerningham Court, James Park Upper Scarborough, Scarborough, Trinidad and Tobago; victorwheeler74@gmail.com

**Keywords:** microbiome, body fat distribution, adiposity, obesity

## Abstract

Objective: The gut microbiome has been associated with visceral fat (VAT) in European and Asian populations; however, associations with VAT and with ectopic fats among African-ancestry individuals are not known. Our objective was to investigate cross-sectional associations of fecal microbiota diversity and composition with VAT and ectopic fat, as well as body mass index (BMI), among middle-aged and older African Caribbean men. Methods: We included in our analysis *n* = 193 men (mean age = 62.2 ± 7.6 years; mean BMI = 28.3 ± 4.9 kg/m^2^) from the Tobago Health Study. We assessed fecal microbiota using V4 16s rRNA gene sequencing. We evaluated multivariable-adjusted associations of microbiota features (alpha diversity, beta diversity, microbiota differential abundance) with BMI and with computed tomography-measured VAT and ectopic fats (pericardial and intermuscular fat; muscle and liver attenuation). Results: Lower alpha diversity was associated with higher VAT and BMI, and somewhat with higher pericardial and liver fat. VAT, BMI, and pericardial fat each explained similar levels of variance in beta diversity. Gram-negative *Prevotellaceae* and *Negativicutes* microbiota showed positive associations, while gram-positive *Ruminococcaceae* microbiota showed inverse associations, with ectopic fats. Conclusions: Fecal microbiota features associated with measures of general adiposity also extend to metabolically pernicious VAT and ectopic fat accumulation in older African-ancestry men.

## 1. Introduction

Visceral fat (VAT) and ectopic fat accumulation, characterized by the deposition of lipids in non-traditional adipose tissues (such as intermuscular fat [IMAT]) and in the liver, heart, and skeletal muscle, can contribute to impaired tissue function, altered tissue metabolism, inflammation, cardiometabolic disease risk, and mortality [1,2,3]. These unhealthy fat depots are known to differ by race, with Black or African-ancestry adults tending to have less VAT [4,5] and liver fat [6] but more IMAT [7,8] than White or European-ancestry adults at a similar BMI.

Several lines of evidence implicate the gut microbiome in the etiology of obesity. For example, in germ-free mice, human intestinal microbiota can cause obesity [9], and in humans, alterations in the gut microbiome have been associated with obesity [10,11]. Moreover, we and others have found that gut microbiome features (e.g., diversity, specific taxa) relate to trunk (i.e., central) fat mass similar to overall fat mass [12,13,14]. However, whether these findings extend to other types of fat accumulation (e.g., VAT, IMAT, liver, pericardial) is not well known. Prior research also indicates that the gut microbiome differs by race and ethnicity [15,16,17], and that race and/or ethnicity may modify associations of the gut microbiome with obesity [18,19] and the distribution of fat [20]. For example, we recently found that the association of gut microbiome alpha diversity with trunk fat differed significantly in Black vs. White adults from the Baltimore Longitudinal Study of Aging [12]. This finding raised several questions about how the gut microbiome relates to fat accumulation in Black populations, in whom there is a paucity of microbiome research.

In the current study, we aimed to address this gap in research in a non-US African-ancestry population that has a different distribution of confounding variables compared to US Black populations. Specifically, our objective was to examine associations of gut microbiota diversity and composition with VAT and ectopic fat volumes (e.g., pericardial, IMAT) and intra-organ lipid storage (e.g., liver attenuation, muscle attenuation) in a well-characterized community cohort of African-ancestry Caribbean men who have low levels of smoking and alcohol intake and non-African genetic admixture [21,22]. We hypothesized that microbiota features associated with VAT volume would be similarly associated with measures of ectopic fat depots and overall fat mass.

## 2. Methods

### 2.1. Study Population

The current analysis was based on a subset of men from the larger Tobago Health Study cohort [23] (see study flowchart in Appendix A). Briefly, the Tobago Health Study began as a population-based prostate cancer screening study from the Caribbean Island of Tobago, Republic of Trinidad and Tobago. Eligible participants were community-dwelling men aged 40+ years who were ambulatory and not terminally ill. The initial study visit occurred from 1997–2003 and recruited 3170 men. Participants were invited to attend follow-up visits (years 2004–2007; 2010–2014), in which body composition measures were obtained. From 2014–2018, we invited a convenience subsample of *n* = 856 existing participants to receive computed tomography (CT) scans of the chest, abdomen, and thigh for ectopic fat assessment. Clinical and lifestyle characteristics were also obtained in this ancillary study.

Beginning in June 2017, participants who had completed CT visits were re-contacted to participate in a microbiome study. Participants were contacted by phone, with emphasis placed on the recruitment of those who most recently completed a CT visit. A convenience sample of 262 men returned to the clinic for additional interview and sample collection; a comparison of the demographic and lifestyle characteristics of these men compared to men who did not return to the clinic is presented in Appendix A. Of these returning men, 259 men donated a fecal sample, and 252 of these samples were able to undergo DNA extraction. For the current analysis, we excluded participants who did not self-identify as African Caribbean (*n* = 25), participants missing CT scans (*n* = 23) or covariate data (*n* = 6), and participants who were using antibiotics within the two weeks prior to fecal sample collection (*n* = 5), resulting in *n* = 193 participants. The time difference between the clinic visit date and the microbiome study date was a median of 2.5 years, with a minimum of 1.1 years and a maximum of 3.4 years.

The Institutional Review Boards of the University of Pittsburgh and the Tobago Ministry of Health and Social Services approved this study. All participants provided written informed consent before data collection.

### 2.2. Fecal Sample Collection and Processing

Participants were given a Zymo Research DNA/RNA Shield Fecal Collection Tube (Zymo Research, Irvine, CA, USA, catalog No. R1100-9-T) at the initial microbiome study interview. Participants self-collected 1 spoonful of feces at home according to manufacturer instructions and refrigerated the samples until they could return the samples to the Calder Hall medical clinic (Tobago). Specimens were stored in the clinic at −80 °C and then shipped on dry ice to the University of Pittsburgh where they continued to be stored at −80 °C. Samples were thawed and aliquoted into 1.5 mL tubes and again stored at −80 °C at the University of Pittsburgh Center for Medicine and the Microbiome.

We performed microbial DNA extraction and sequencing at the University of Pittsburgh Center for Medicine and the Microbiome. We extracted microbial DNA using the Qiagen PowerSoil DNA Isolation kits (MO BIO Laboratories, Carlsbad, CA, USA). We performed PCR using barcoded amplicons of the 16S V4 rRNA gene variable region with primers 515F 5′-(GTG CCA GCM GCC GCG GTA A)-3′ and 806R 5′-(GGA CTA CHV GGG TWT CTA AT)-3′. We purified samples using magnetic bead size selection (AMPure XP, Beckman Coulter, Brea, CA, USA) and pooled them for sequencing (Illumina MiSeq; Illumina, San Diego, CA, USA).

We demultiplexed reads using standard Illumina software (MiSeq Software Version: 4.1.0.656). We assessed the quality control of reads using an in-house software pipeline that performs dust low complexity filtering, quality value trimming, primer trimming, and minimum read length filtering. We merged the forward and reverse reads passing quality control thresholds into contigs, and then processed and taxonomically classified them using an in-house Mothur [24]-dependent pipeline. We classified sequences using the Ribosomal Database Project’s (RDP) naïve Bayesian classifier [25,26] and the SILVA 16S rRNA database (v138) [25].

### 2.3. Computed Tomography Assessments

We collected CT scans of the chest, abdomen, and thighs at the Calder Hall Medical Clinic, Tobago, using a GE dual slice, high-speed NX/I CT scanner (GE Medical Systems, Waukesha, WI, USA). Scan slices were 3 mm thick with a 500 mm display field of view, and the scanner settings were 120 KVp, 250 mA (or increased to 300 mA if participants were >200 lbs.), 0.7 s gantry speed, and pitch of 1.5:1. A single individual collected the scans for all participants using the same CT scanner, and CT contrast was not used. We electronically transmitted scans to the central CT reading center at Vanderbilt University Medical Center for image analysis.

We analyzed images using previously described methods [27,28,29]. Briefly, a radiologist-trained analyst used a dedicated imaging processing workstation with custom-programmed subroutines (OsiriX, Pixmeo, Geneva, Switzerland) and a dedicated pen computing display (Cintiq, Wacom Technology Corporation, Vancouver, WA, USA) to manually trace tissue and anatomical boundaries in CT scans. We defined adipose tissue as voxels within −190 to −30 Hounsfield Units (HU) and lean muscle tissue as voxels within −29 to 160 HU. We calculated the volume (cm^3^) of each tissue. We defined muscle attenuation as the average HU of the lean muscle tissue. We provide details on VAT and specific ectopic fat locations and measurements below, including pericardial fat, paraspinous IMAT and muscle attenuation, psoas IMAT and muscle attenuation, thigh IMAT and muscle attenuation, and liver attenuation.

Pericardial fat: At the level of the left main artery, an index slice was identified and slices within 15 mm above and 30 mm below the index slice were selected for the measurement of adipose tissue. We defined pericardial fat as the adipose tissue located in the membranes surrounding the heart and the roots of major blood vessels, aggregating the tissue-containing pixels and accounting for slice thickness [30].

VAT: We centered 3 contiguous slices at L4–L5 using a lateral scout image to identify the *z*-axis location of the L4–L5 intervertebral space. Scans included the midpoint and the slices immediately above and below that point. We defined VAT as the adipose tissue located within the peritoneal cavity.

Paraspinous and psoas IMAT and muscle attenuation: We centered 3 contiguous slices at L3–L4 using a lateral scout image to identify the *z*-axis location of the L3–L4 intervertebral space. Scans included the midpoint and the slices immediately above and below that point. Our trained analyst traced boundaries at the paraspinous and psoas muscles and fascia. We defined IMAT as the sum of adipose tissues located within paraspinous or psoas muscle groups across both sides of the body and muscle attenuation as the average attenuation of paraspinous or psoas muscle groups across both sides of the body.

Thigh IMAT and muscle attenuation: We centered 10 contiguous slices at the mid-thigh level in both legs (determined using an anterior–posterior scout scan of the entire femur). The slices were comprised of the midthigh point, the 4 slices above, and the 5 slices below that location. Our trained analyst traced boundaries at the thigh muscles and fascia in 3 of the 10 slices, and these were imputed over the remaining slices (with the analyst verifying imputation accuracy). We defined thigh muscle as the sum of the adductors, hamstrings, and quadriceps muscles; we defined thigh IMAT as the sum of adipose tissues located within thigh muscle groups across both thighs; and we defined thigh muscle attenuation as the average attenuation of thigh muscle groups across both thighs.

Liver attenuation: We centered 3 contiguous slices at T12–L1. We measured 3 regions of interest in each slice and defined mean liver attenuation as the average attenuation of these 9 regions.

### 2.4. Other Variables

At the 2004–2007 baseline body composition visit, we collected education status (categorized as “Primary” (1st–5th Standard, ages 5–11), “Secondary” (high school, Form 1st thru 6th, ages 11–18), or “post-Secondary” (any education or training beyond secondary)) using interviewer-administered questionnaires. Given that all participants were 40 years or older at this visit, we assumed that the educational attainment they reported was their highest educational attainment.

At the most recent clinic visit (2014–2018), we assessed age (years), current smoking status [yes/no], self-reported alcohol intake of 4 or more drinks per week [yes/no], and self-reported hours walked for physical activity (hours/week) using standardized interviewer-administered questionnaires. We measured height to the nearest 0.1 cm using a wall-mounted stadiometer and we measured body weight to the nearest 0.1 kg without shoes using a balance beam scale. We calculated BMI from body weight and standing height (kg/m^2^). At this visit, we collected dietary intake using a 146-item semi-quantitative monthly food frequency questionnaire tailored to the Trinidad and Tobago population [31]. Participants missing ≥10% of food items or reporting extreme energy intakes (<600 kcal/day or >5000 kcal/day) were excluded from analyses that included diet as a covariate. We used all available valid participant dietary data from this visit (*n* = 798) to construct a population-level alternative Mediterranean diet score, based on the modified Mediterranean diet score by Fung et al. [32]. Briefly, participants receive a positive point for having above-population-median intakes of vegetables, fruits, nuts, whole grains, legumes, fish, and ratios of monounsaturated fat to saturated fat; a positive point for having an alcohol intake between 5–15 g ethanol/day; and a positive point for having below-population-median intakes of red and processed meats. The sum of points creates a score ranging from 0 to 9, with 9 indicating the highest adherence to the alternative Mediterranean-style dietary pattern. Of the *n* = 193 participants in our analytic sample, there were *n* = 164 who had valid dietary data and an assigned alternative Mediterranean-style dietary score.

At the fecal collection visit (2017–2018), we asked participants to bring all medications taken in the prior month, including antibiotics used within the prior two weeks. We categorized antibiotic use at the time of the fecal collection visit as yes/no.

### 2.5. Statistical Methods

Statistical analyses were performed using R statistical software (version 4.3.1). We present sample characteristics as either mean (standard deviation), median (interquartile range), or *n* (%). We assessed age-adjusted partial Pearson correlations between BMI and ectopic fat measures.

#### 2.5.1. Diversity Analyses

We calculated microbiota alpha diversity measurements based on rarefied datasets. Microbiota data from our overall analytic sample had a minimum sample depth of 1516 and 347 unique OTUs. We rarefied samples ten times without replacement to a depth of 1500. We rounded the average of these results to the nearest integer, resulting in the removal of 128 OTUs and a final table of 219 unique OTUs.

We calculated alpha diversity metrics (observed OTUs, Pielou’s evenness, and Shannon index) using the R package phyloseq (version 1.44.0) [33]. We constructed linear regression models with an ectopic fat measure as the dependent variable and an alpha diversity metric as the independent variable, and we analyzed models separately for each ectopic fat measure and for each alpha diversity metric. We analyzed models in 2 stages:Model 1: unadjusted (with the exception that models with observed OTUs as the independent variable additionally adjusted for unrarefied sequencing depth);Model 2: Model 1 + adjustment for age (years), education status (categorical), current smoking status (yes/no), drinking 4+ alcoholic beverages per week (yes/no), hours walked per week for exercise (hours), and the time difference between CT scan and fecal sample collection (years).

We calculated microbiota beta diversity using the Bray-Curtis distance from the phyloseq package. We tested for associations with beta diversity with permutational analysis of variance (PERMANOVA) models using the adonis2 function in the vegan (version 2.6-4) [34] package with 9999 permutations. We included an ectopic fat measure as the independent variable and adjusted for Model 2 covariates as indicated above (age, education status, current smoking status, drinking 4+ alcoholic drinks per week, hours walked per week, and time difference between CT scan and fecal sample collection). We analyzed models separately for each ectopic fat measure.

#### 2.5.2. Compositional Analyses

We removed rare (prevalence of <10%) and low-abundant (mean relative abundance < 0.1%) OTUs from the non-rarefied OTU tables. These filtering criteria reduced the number of unique OTUs by 151 and 73, respectively, resulting in 78 OTUs for differential abundance analysis. We next performed differential abundance testing using the Analysis of Compositions of Microbiomes with Bias Correction (ANCOM-BC) [35] method. Briefly, ANCOM-BC uses a linear regression framework with a sampling bias offset term to identify associations of variables of interest with the differential absolute abundance of taxa. We included an ectopic fat measure as the dependent variable and adjusted for Model 2 covariates listed in diversity analyses (age, education status, current smoking status, drinking 4+ alcoholic drinks per week, hours walked per week, and time difference between CT scan and fecal sample collection). We analyzed models separately for each ectopic fat measure. We used ANCOM-BC version 2.2.2 with default parameters (except for setting the prevalence filtering to 0 since we pre-filtered our microbial count table). We used an FDR-corrected q-value of <0.05 to indicate OTUs significantly associated with ectopic fat measures. We visualized associations of OTUs that were statistically significantly associated with ectopic fat measures using heat maps of standardized model coefficients.

#### 2.5.3. Sensitivity Analyses

We performed a variety of sensitivity analyses to evaluate the robustness of our findings, including (1) testing effect measure modification by age in alpha diversity models, (2) assessing potential confounding by muscle volume on muscle fat measures, (3) assessing potential confounding due to dietary intake, and (4) testing the robustness of differential abundance results using different statistical methods.

To evaluate effect measure modification by age in alpha diversity models, we tested the significance of a cross-product interaction term between continuous age and microbiome alpha diversity in different ectopic fat models. We used Johnson–Neyman plots to identify values of age where the effect of alpha diversity significantly differed from 0.

To determine the potential confounding effects of muscle volume on muscle fat (pericardial, IMAT, and muscle attenuation) associations with microbiota diversity, we compared the fully adjusted models with and without additional adjustment for respective muscle volumes (cardiac, psoas, paraspinous, or thigh). To determine the potential confounding effects of dietary intake on diversity associations, we re-analyzed alpha diversity and beta diversity models in the subset of individuals with valid dietary data (*n* = 164) and compared the change in model parameters before and after adjustment for total energy intake and Alternative Mediterranean-style diet score. To further understand the interplay between diet, gut microbiota, and BMI and fat measures, we also include an analysis of the associations of total energy intake and Alternative Mediterranean-style diet score with BMI, fat measures, and alpha diversity measures.

To corroborate our primary ANCOM-BC differential abundance testing results, we also analyzed differential abundance using the logistic compositional analysis (LOCOM) [36] and the ALDEx2 [37] methods. Briefly, LOCOM models associations of variables of interest with microbial differential abundance through a robust logistic regression approach. We used LOCOM version 1.1 with default parameters (except for setting the prevalence filtering to 0 since we pre-filtered our microbial count table). ALDEx2 infers taxon abundances from observed counts by drawing Monte-Carlo instances from a Dirichlet distribution and then regresses the center log-ratio transform of these abundances onto the variables of interest. We used ALDEx2 version 1.32.0 with default parameters. For both LOCOM and ALDEx2, model independent variables were identical to those in the ANCOM-BC modeling approach, and we used an FDR-corrected q-value of <0.20 to indicate OTUs significantly associated with ectopic fat measures.

## 3. Results

### 3.1. Sample Characteristics

We report demographic, lifestyle, ectopic fat, and microbiota alpha diversity metric characteristics in Table 1, overall and by the quartile of VAT. Participants were predominantly middle-aged or older, were generally non-smokers, and on average had an overweight BMI. BMI and ectopic fat volumes increased, while muscle and liver attenuations decreased, across increasing quartiles of VAT.

We present inter-variable correlations for BMI and fat measures in Appendix A. Briefly, we identified moderate to strong positive intercorrelations (r = 0.37–0.75) among BMI, VAT, and ectopic fat volumes, as well as moderate inverse correlations (r = −0.19–−0.51) of these measures with liver attenuation. Muscle attenuation measures were more strongly correlated with each other as well as with their respective IMAT volumes.

### 3.2. Microbiota Characteristics

We averaged 11,379 16S rRNA gene sequence reads per sample. We identified 18 phyla, 111 families, and 347 OTUs. On average, participants had 60 unique OTUs per fecal sample. We present the fecal microbiota compositions, at the phylum and genus levels, by quartile of VAT in Figure 1. We found higher abundances of microbes from the *Bacteroidetes* phylum, with the OTU *Prevotella_9* having a mean relative abundance of ~31% across samples. *Bacteroidetes* members including *Prevotella_9* seemed to increase in relative abundance with increasing quartiles of VAT. We also report that microbiota alpha diversity measures of Pielou’s evenness and Shannon diversity also decreased across increasing quartiles of VAT (Table 1).

### 3.3. Diversity Analyses

We present linear regression model results for the associations of BMI, VAT, and ectopic fats with alpha diversity metrics in Table 2. There were no statistically significant associations of BMI, VAT, or ectopic fat measures with microbiota richness (estimated by the unique number of observed OTUs). However, both BMI and VAT were significantly and inversely associated with Pielou’s evenness and Shannon Diversity metrics in covariate-adjusted models. Moreover, while not statistically significant, higher levels of pericardial fat and lower levels of liver attenuation (i.e., more hepatic lipid accumulation) were also associated with lower Pielou’s evenness and Shannon Diversity metrics.

We depict Bray-Curtis dissimilarity principal coordinate analysis plots by quartiles of BMI and VAT in Figure 2. BMI and VAT were each significantly associated with Bray-Curtis dissimilarity in PERMANOVA models after covariate adjustment (Table 3), with each measure explaining roughly 1.9% of the variance in the Bray-Curtis dissimilarity. Other fat measures were not significantly associated with Bray-Curtis dissimilarity (Table 3).

### 3.4. Compositional Analyses

We measured the associations of microbial OTUs with a per standard deviation increment in BMI, VAT, or ectopic fat measure using the ANCOM-BC; the results are visualized in Figure 3. We identified 13 bacterial OTUs that were significantly differentially associated with at least one BMI, VAT, or ectopic fat measure after FDR correction. VAT had the greatest number of significantly associated taxa, with eight OTUs being positively associated (predominantly Gram-negative *Bacteroides* bacteria from the *Prevotellaceae* family and Gram-negative Firmicutes bacteria from the *Selenomonodales* order) and one OTU having an inverse association (*Ruminococcus_2*). While there were fewer significant associations of OTUs with BMI and ectopic fat measures, the associations were generally of similar magnitude and direction as seen with VAT.

### 3.5. Sensitivity Analyses

We identified significant effect modification by age for psoas IMAT and for thigh muscle attenuation in observed OTU models, and for psoas muscle attenuation in Pielou’s evenness models (all interaction *p* > 0.05; *p*-values presented in Appendix A). We visually explored these interactions using Johnson–Neyman plots (not presented) and found that the effect of alpha diversity metrics on those ectopic fat measures was only significant in a few participants (*n* ≤ 14) who were aged 75 or older; thus, caution is warranted in further interpreting these findings.

For muscle fat measures, we investigated the potential confounding of microbiota diversity associations by muscle volume (i.e., cardiac, psoas, paraspinous, or thigh muscle volumes), and for all BMI and fat measures, we investigated potential confounding by diet (i.e., total energy intake and Alternative Mediterranean Diet adherence score in the *n* = 164 participants with diet data). Adjustment for muscle volumes (Appendix A) or dietary data (Appendix A) did not appreciably change fat associations with alpha or beta diversity measures, respectively.

To gain a better understanding of the interplay between diet, gut microbiota, and obesity or fat measures in this population, we report the associations of the Alternative Mediterranean Diet score and of total energy intake with BMI, fat measures, and alpha diversity measures (Appendix A). Greater adherence to the Alternative Mediterranean Diet score was associated with lower liver attenuation, representing increased liver fat accumulation, and with lower alpha diversity measures (albeit not statistically significantly). In contrast, total energy intake was not statistically significantly associated with any BMI, fat measure, or alpha diversity measure.

We reanalyzed differential abundance associations using the ALDEx2 and LOCOM methods to corroborate our ANCOM-BC findings (Appendix A). While the magnitudes of associations are not able to be directly compared across approaches, the directions of associations were largely consistent across methods, suggesting that each modeling approach agreed on whether an OTU was generally positively or negatively associated with an obesity or fat measure of interest.

## 4. Discussion

In this cohort of African-ancestry men from the Caribbean, we found that lower fecal microbiota diversity and overall microbiota composition (i.e., beta diversity) were associated with higher levels of BMI and VAT. While associations with ectopic fat depots were less marked, directions of association were similar. Our findings show that while fecal microbiota diversity may be more strongly associated with BMI and VAT than with ectopic fat depots, specific microbial OTUs may track ectopic lipid accumulation across body sites and tissue types. Thus, interventions targeting gut microbes may influence multiple ectopic fat depots across different anatomical sites, with perhaps a more pronounced effect on VAT.

### 4.1. The Importance of Race/Ethnicity and Geography in Microbiota–Obesity Relationships

Recent studies point to the importance of including diverse populations in microbiome and obesity research. Research in US populations has found that associations of some microbiome features with measures of obesity, trunk fat [18], and liver fat [20] differ by racial identification. Additionally, within individuals of African ancestry, the associations of the microbiome and microbially produced short-chain fatty acids with obesity differed by where the study country was along the epidemiologic transition [19]. Such findings indicate that understanding the relationship between gut microbiota and health, and its applications to addressing health disparities, depends on further social and demographic contexts and can be instrumental in informing population-specific interventions.

Microbiome–obesity research in Caribbean populations is sparse [19,38]. Our study is the first to report fecal microbiota characteristics from the Caribbean Island of Tobago, Trinidad and Tobago. To the best of our knowledge, only one other study has reported fecal microbiota characteristics from this country, specifically from the island of Trinidad [38]. It is also important to note that Trinidad differs considerably in racial/ethnic composition (i.e., much larger West Asian ancestry) compared to Tobago (i.e., much larger African ancestry). We report a high prevalence of *Prevotella* genera in our study cohort, often shown to be in higher abundance in populations with high-fiber diets [39]. In contrast, *Prevotella* were not dominant in Jamaican or Trinidadian studies [19,38]. Further profiling of the human microbiome across the ethnically and culturally diverse Caribbean and research into factor(s) driving differences in microbiome profiles across Caribbean islands is needed.

### 4.2. Fecal Microbiota Diversity Is Associated with Some, but Not All, Ectopic Fats

Our findings of inverse relationships of fecal microbiota alpha diversity with BMI, VAT, and somewhat with liver fat are mostly supported in the literature. The relationship between alpha diversity measures and obesity is somewhat inconsistent, as previous meta-analyses [40,41] have reported either small or insignificant inverse associations of alpha diversity with obesity. Additionally, studies have also reported inverse associations of alpha diversity with measures of VAT [14,42] and that individuals with metabolic dysfunction-associated fatty liver disease have lower alpha diversity than controls [20,43,44]. Given that studies generally report consistency in associations of microbiome features across measures of overall and central adiposity [12,13,14] and the high correlations between BMI, VAT, and liver fat, these findings are not unexpected. While we did report associations of beta diversity with BMI and VAT, we did not find that these associations extended to liver fat, in contrast to prior literature [20,43,44]. This may be due in part to these studies comparing individuals based on the presence of having fatty liver disease. In our study cohort, liver attenuation was relatively high (i.e., low liver-fat accumulation), with only *n* = 9 of our participants reaching thresholds suggesting the presence of a fatty liver disease. Thus, it is possible that larger variations in liver fat are needed to see associations with global microbiota structure.

Muscle fat infiltration, also called myosteatosis, is thought to play a role in metabolism, cardiovascular disease, and physical functioning [45]. However, there is sparse information on associations of gut microbiota with measures of myosteatosis. To the best of our knowledge, we believe our study is the first in humans to investigate associations of fecal microbiota features with pericardial fat and with intermuscular fat, and one of the few [46] to investigate associations with muscle attenuation, an indirect marker of intramuscular fat. While it did not reach statistical significance, we identified an inverse association between some alpha diversity metrics (Pielou’s evenness, Shannon index) with pericardial fat and reported that pericardial fat accounted for the third most variance in Bray-Curtis dissimilarity, following BMI and VAT. Pericardial fat is highly correlated with BMI and VAT, and its associations with metabolic health are attenuated after adjustment for VAT, suggesting it may be a marker for VAT [47]. This could in part explain the shared inverse association between microbiota alpha diversity and both VAT and pericardial fat. However, we did not identify any associations of alpha or beta diversity with IMAT or with muscle attenuation. This is somewhat surprising, given the intercorrelations of ectopic fat depots in this study. Additional studies investigating microbiota diversity associations with myosteatosis in humans are needed to confirm these findings.

### 4.3. Fecal Microbiota Taxa Are Similarly Associated with Overall Obesity and Ectopic Fat Accumulation

We report several microbial OTUs that were associated with BMI, VAT, and ectopic fats. Importantly, while the magnitude of association differed by fat measures, the directions of association were fairly consistent, again in line with prior findings of consistency in associations across obesity measures [12,13,14]. In general, we report that OTUs associated with the *Prevotellaceae* family and *Negativicutes* class (i.e., *Acidaminococcus* and *Megamonas*) had positive associations with BMI, VAT, or ectopic fat depots, while *Ruminococcaceae* OTUs (i.e., *Ruminoclostridium_9* and *Ruminococcus_2*) were generally inversely associated with these measures.

Of the ectopic fat depots included in our analysis, the associations of gut microbiota composition with liver fat have been most widely studied. Fecal microbiota transfer from humans with nonalcoholic steatohepatitis to germ-free mice can cause increases in liver triglycerides [48,49]. Further, microbial composition alterations may precede clinical fatty liver development [50], suggesting this is not merely a consequence of the condition. However, most microbial taxonomic associations with liver fat have not been consistent across studies [51,52]. While it is hypothesized that factors such as differences in study design and confounder control contribute to these discrepancies, it is also suggested that differences in geography and in cohort racial and ethnic makeup are significant contributors to this variation and that there may be multiple microbial compositions that may promote liver fat accumulation via different mechanisms [51]. Thus, the inclusion of geographically and ethnically diverse populations is needed to better understand the features and mechanisms underlying fat accumulation.

To the best of our knowledge, only one other study has reported on microbiota associations with a measure of skeletal muscle adiposity (among *n* = 37 adults with obesity) [46]. This study identified several taxa associated with muscle attenuation, including taxa that were associated with lower muscle attenuation (e.g., members from the *Ruminococcaceae* and *Lachnospiraceae* families) and associated with higher muscle attenuation (e.g., *Clostridiaceae* and *Clostridium sensu stricto*). Our findings diverge from this, as we report a positive association of a *Lachnospiraceae* OTU and a negative association of an unclassified *Bacteroidales* OTU with both psoas and thigh muscle attenuations. Larger and more diverse cohorts are needed to further elucidate the relationship of gut microbiota with measures of myosteatosis.

*Prevotella* are fiber-degrading bacteria [53] and are thought to track with higher-fiber diets; however, they have also been positively associated with obesity [41,54]. It has been hypothesized that the discrepancy in whether *Prevotella* associates with health or disease could be due to strain diversity within *Prevotella* species [55] or perhaps even its co-occurrence with other obesogenic taxa [54]. The *Negativicutes* genera *Acidaminococcus* has also been associated with obesity [41,54], and *Megamonas* was reported to be higher in individuals with obesity [56] and with non-alcoholic fatty liver [20,56]. Notably, *Prevotellaceae* and *Negativicutes* bacteria are all gram-negative bacteria capable of producing pro-inflammatory lipopolysaccharides. Lipopolysaccharide-induced inflammation is hypothesized to be one mechanism linking intestinal microbiota to metabolic health [57], and we have previously shown in our cohort that a surrogate biomarker of lipopolysaccharide-induced inflammation was associated with increases in central obesity and muscle fat accumulation [58]. Thus, it is possible that higher levels of these gram-negative taxa could contribute to worsening ectopic fat accumulation via the production of lipopolysaccharides.

The remaining OTUs we identified as being associated with fat measures in our study belonged to the Clostridiales order, including members of the *Ruminococcaceae* family (i.e., *Ruminoclostridium_9* and *Ruminococcus_2*) and the *Lachnospiraceae* family (Lachnospiraceae_Incertae_Sedis, Lachnoclostridium, and Roseburia). These families contain known producers of short-chain fatty acids, microbial metabolites that may have beneficial influences on host metabolism [59]. However, our findings were mixed, with *Ruminococcaceae* OTUs being inversely associated, and *Lachnospiraceae* OTUs showing both positive and negative associations with fat measures. One *Lachnospiraceae* member, *Lachnoclostridium*, was found to be enriched in individuals with non-alcoholic fatty liver disease and with liver fat accumulation in a multi-ethnic US cohort [20,60], though an ethnicity-stratified analysis found that this relationship remained significant only for Hispanic individuals [20]. We also report an inverse association of *Lachnoclostridium* with liver attenuation, corroborating the finding that this taxon is associated with liver fat accumulation. Members of the *Lachnoclostridium* genus have the ability to produce trimethylamine from dietary choline [61], which is absorbed and further metabolized by the liver into trimethylamine N-oxide (TMAO). TMAO is associated with increased cardiovascular disease risk [62,63,64] and was shown to be positively correlated with non-alcoholic fatty liver disease severity [65]. It is hypothesized that TMAO can influence liver fat accumulation through the depletion of absorbed choline levels (via its conversion to trimethylamine) or through influencing bile acid metabolism [52].

### 4.4. Perspectives and Implications for Future Research

Our study findings have implications for future analyses of gut microbes with overall adiposity and body composition. Our current findings and the prior supporting literature [12,13,14] reinforce the notion that measures of fecal microbiota diversity and composition are associated with regional or tissue-specific measures of fat accumulation in a similar direction and magnitude as they are associated with measures of overall obesity. This suggests that inferences made from fecal microbiota diversity and composition associations with BMI could be reasonably extended to ectopic fat measures without the need for direct imaging methods, which can be expensive and impractical in large epidemiological settings. It may also suggest that targeting obesogenic features of the gut microbiome through microbiota-based interventions could have similarly beneficial impacts across body regions and tissues.

Various mechanisms linking the gut microbiome to obesity have been suggested, including influencing the amount of energy harvested from the diet, promoting gut permeability and subsequent inflammation, and altering host metabolism through the production of metabolites [57]. Whether microbiota-focused interventions should target microbial compositions or microbial metabolites remains to be determined. Thus, future investigations including other microbiota features (e.g., microbial metabolites) are also needed to uncover microbial targets for obesity interventions and to help uncover the mechanisms underpinning microbiota associations with obesity.

Though the microbiome features within a population may be similarly associated with obesity phenotypes, prior research also suggests these features may not always have the same associations in other populations or within population subgroups (such as other racial and ethnic groups) [12,18,19,20]. This has implications for the development and implementation of targeted microbiota-based obesity interventions. The inclusion of diverse participants in microbiome research is thus needed to determine if there are specific microbiome features (e.g., microbial diversity, specific microbe(s), or microbial genes and metabolites) that contribute to obesity across population groups—or, conversely, to determine if there are no “universal” features.

Given the many ways in which microbes can contribute to the development of obesity and fat accumulation [57], it is likely that there are also multiple variations in gut microbiota features that contribute to obesity. The reasons for racial and ethnic differences in the gut microbiome are likely multifactorial. Heritable taxa such as *Odoribacter* and *Christensenellaceae* showed different associations with ethnic groups in US-based studies [16], suggesting some role for genetic ancestry in contributing to ethnic differences in the microbiome. However, the differences most likely derive from environmental (i.e., non-genetic) factors, including variations in diet, culture, socioeconomic factors, and geography [66,67,68,69,70,71,72]. It may be possible that the environmental factors that drive racial and ethnic differences in the gut microbiome may also differentially promote different kinds of obesogenic microbiota features.

While the current study was not designed to investigate racial or ethnic differences, it does provide data on microbiota and adiposity relationships in an understudied and high-metabolic-disease-risk African-ancestry group, in a setting with a unique confounding structure (e.g., low rates of alcohol and smoking, low non-African genetic admixture) compared to other African-ancestry groups (e.g., US African Americans). There are likely cultural differences including the degree of Westernization between the sub-Saharan Africans, U.K. Africans, African Caribbeans, and African Americans, as well as between various Caribbean islands. These cultural and geographic differences may contribute to differences in microbiota compositions between these groups. Thus, future microbiome research should include populations of African ancestry living outside of the Caribbean region, as well as on other Caribbean islands with predominantly African Caribbeans.

### 4.5. Study Limitations

Our study has potential limitations. Our analysis is cross-sectional, so we cannot assess temporality or causality between microbiota features and fat accumulation, and our findings may be impacted by unmeasured confounders, including pre-existing health conditions that may influence a participant’s lifestyle factors. While we adjusted for various lifestyle factors in our analysis, our physical activity data were based on self-report (which may not be as accurate as objectively measured physical activity), and dietary data were only available in a subset of participants, thus limiting a more detailed analysis of the impact of diet as a confounder. Additionally, our fecal samples were collected up to 3 years following the CT scan measurements; while we cannot rule out potential changes in lifestyle that may have occurred between the CT scan visit and fecal sample collection, we did adjust for the time difference between visits and for several lifestyle factors at the CT scan visit. Because our study cohort is restricted to middle-aged and older men of African ancestry from the Caribbean, findings may not be generalizable to females or younger-aged men or to non-African Caribbean populations. Further, participants in the microbiome ancillary study were invited back to contribute fecal samples after having completed the CT study visit in the parent cohort; thus, there is the potential for self-selection bias to exist. While a comparison of demographic and lifestyle characteristics suggests that the microbiome subsample was, on average, 3 years younger than the larger cohort, we acknowledge the potential for these participants to be different by other unmeasured characteristics. Future studies could improve upon our findings by avoiding these limitations; for example, through longitudinal study designs, more detailed data collection for lifestyle characteristics, and using random sampling methods if collecting fecal samples in a subset of study participants to reduce self-selection bias.

## 5. Conclusions

We report, in a cohort of middle-aged and older African-ancestry men from the Caribbean, that lower fecal microbiota diversity and alterations in fecal microbiota composition (e.g., higher *Prevotella* and *Negativicutes*, lower *Ruminococcaceae*) were associated with higher BMI and VAT, and these microbiota features were also associated, albeit to a lesser extent, with greater ectopic fat deposition across multiple body tissues. Taken together with previous literature, our findings show that fecal microbiota features associated with general obesity also extend to metabolically pernicious fat accumulation. Future interventional studies are needed to determine if modulating these microbiota features can reduce ectopic fat accumulation and improve cardiometabolic health.

## Figures and Tables

**Figure 1 microorganisms-12-00812-f001:**
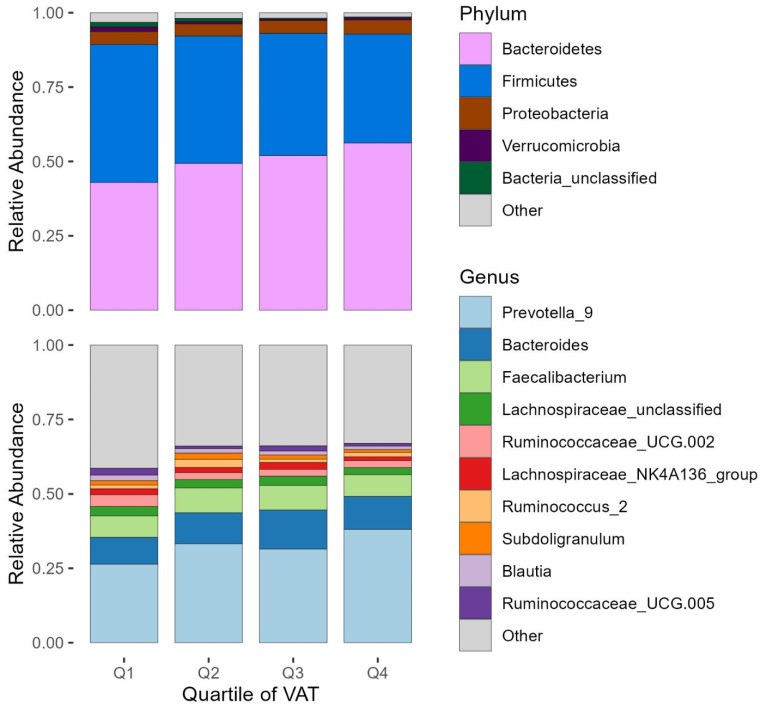
Relative abundance plots of fecal microbiota by quartile of VAT among men from the Tobago Health Study. A total of 193 men were included in the analyses. Participants were grouped into quartiles of VAT. The top 5 most abundant phyla and the top 10 most abundant genera in the overall study sample are shown; the remaining taxa are grouped together into an “Other” category. Abbreviations: VAT = visceral fat.

**Figure 2 microorganisms-12-00812-f002:**
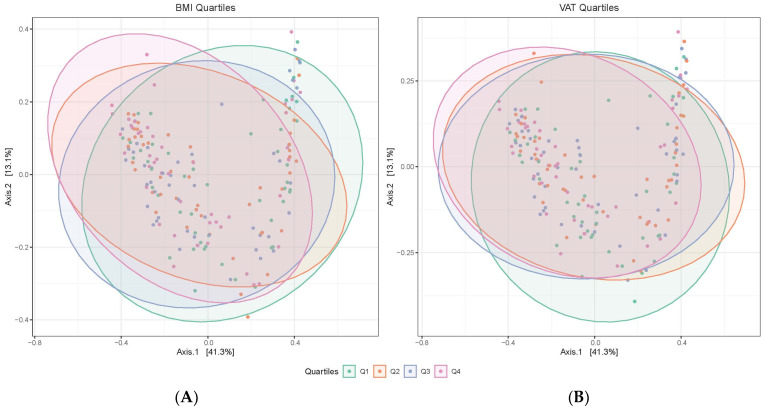
PCoA plots of Bray-Curtis dissimilarity by quartiles of BMI and VAT among men from the Tobago Health Study. A total of *n* = 193 men were included in the analyses. Participants were grouped into quartiles of BMI or VAT. Fecal microbiota samples were first rarefied to a sequencing depth of 1500 reads, and then the Bray-Curtis dissimilarity was calculated for each pair of samples. PCoA plots were constructed to visualize the separation of samples, such that the samples located closer together had more similar microbial compositions. A 95% ellipse was calculated for each BMI or VAT quartile. Panels: (**A**) BMI quartiles; (**B**) VAT quartiles. Abbreviations: PCoA = principal coordinates analysis; VAT = visceral fat.

**Figure 3 microorganisms-12-00812-f003:**
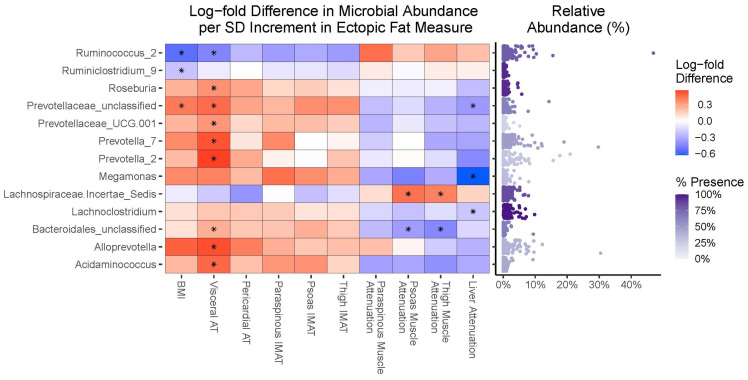
Heatmap and relative abundance plots for associations of microbial OTUs with standard deviation increment in BMI or fat measures. A total of *n* = 193 men were included in the analyses. Rare (prevalence of <10%) and low-abundant (mean relative abundance < 0.1%) microbes were pre-filtered from samples. ANCOM-BC models were analyzed separately for each BMI or fat measure; models adjusted for age (years), educational attainment (primary, secondary, or post-secondary), hours walked per week for exercise (hours), current smoking status (yes vs. no), drinking 4 or more alcoholic drinks per week (yes vs. no), and the time difference between CT scans and fecal sample collection (years). Heatmap (**left**): The log-fold difference in microbial abundance per standard deviation increment in BMI or fat measure. Asterisks (*) indicate significant associations at FDR-corrected q < 0.05. Note: lower muscle attenuation or lower liver attenuation reflects greater lipid accumulation. Relative abundance plot (**right**): Strip plot showing the relative abundance of each OTU. Colors of strip plots reflect the percent non-zero prevalence of the OTU across the analytic sample. Abbreviations: OTU = operational taxonomic unit; ANCOM-BC = Analysis of Compositions of Microbiomes with Bias Correction; FDR = false discovery rate; IMAT = intermuscular fat; SD = standard deviation.

**Table 1 microorganisms-12-00812-t001:** Sample characteristics, overall and by quartile of visceral adipose tissue volume, for *n* = 193 men from the Tobago Cohort Study.

	Overall (*n* = 193)	VAT Q1 (*n* = 49)	VAT Q2 (*n* = 48)	VAT Q3 (*n* = 48)	VAT Q4 (*n* = 48)	Linear Trend *p*-Value
Age (years)	60.0 [56.0, 68.0]	60.0 [56.0, 68.0]	58.5 [55.0, 63.3]	60.0 [55.0, 70.0]	61.0 [57.8, 68.3]	0.280
Education						
Primary	147 (76.2%)	41 (83.7%)	32 (66.7%)	39 (81.3%)	35 (72.9%)	0.072
Secondary	28 (14.5%)	2 (4.1%)	9 (18.8%)	6 (12.5%)	11 (22.9%)	
Post-Secondary	18 (9.3%)	6 (12.2%)	7 (14.6%)	3 (6.3%)	2 (4.2%)	
Hours Walked/Week	2.5 [0.0, 6.0]	2.0 [0.0, 5.3]	2.3 [0.0, 6.3]	2.5 [0.0, 5.3]	2.8 [0.0, 6.0]	0.858
BMI (kg/m^2^)	28.3 (4.9)	24.2 (3.1)	27.7 (3.2)	29.1 (4.0)	32.2 (5.2)	<0.001
Current Smoking Status	18 (9.3%)	9 (18.4%)	5 (10.4%)	2 (4.2%)	2 (4.2%)	0.051
Has 4+ Alcoholic Drinks/Week	25 (13.0%)	4 (8.2%)	7 (14.6%)	7 (14.6%)	7 (14.6%)	0.722
Time Difference between Measures (years)	2.5 [1.6, 2.6]	2.5 [1.6, 2.8]	2.5 [1.7, 2.7]	2.5 [1.7, 2.6]	1.7 [1.6, 2.5]	0.137
Fat Measures						
Abdominal VAT (cm^3^)	93.0 [55.7, 124.6]	39.2 [22.9, 48.7]	70.6 [63.4, 80.6]	111.2 [105.7, 118.1]	150.9 [136.1, 172.3]	<0.001
Pericardial Fat (cm^3^)	30.0 [19.3, 46.4]	18.3 [12.7, 25.7]	26.6 [18.4, 37.5]	34.0 [22.3, 44.6]	48.3 [40.5, 62.7]	<0.001
Psoas IMAT (cm^3^)	0.6 [0.4, 0.9]	0.4 [0.2, 0.6]	0.6 [0.5, 0.8]	0.7 [0.5, 0.9]	0.9 [0.6, 1.2]	<0.001
Paraspinous IMAT (cm^3^)	2.4 [1.7, 3.4]	1.7 [1.0, 2.3]	2.4 [1.8, 2.9]	2.5 [2.1, 3.9]	3.2 [2.4, 4.2]	<0.001
Thigh IMAT (cm^3^)	105.6 [83.8, 134.6]	69.7 [42.3, 95.8]	103.7 [88.2, 128.7]	111.4 [95.6, 131.3]	139.9 [106.9, 182.8]	<0.001
Psoas Muscle Attenuation (HU)	49.2 [46.1, 51.3]	50.9 [49.1, 52.4]	49.6 [47.9, 51.4]	48.5 [45.5, 51.0]	47.1 [44.7, 50.0]	<0.001
Paraspinous Muscle Attenuation (HU)	44.8 [39.9, 48.6]	47.9 [44.8, 50.3]	45.3 [41.9, 48.5]	44.1 [37.4, 48.2]	42.9 [37.3, 45.3]	<0.001
Thigh Muscle Attenuation (HU)	43.9 [41.1, 45.9]	45.4 [43.1, 46.8]	44.8 [42.5, 46.4]	42.8 [40.9, 45.1]	41.3 [39.7, 44.0]	<0.001
Liver Attenuation (HU)	57.5 [53.4, 61.5]	61.3 [58.6, 63.3]	59.1 [55.3, 61.2]	56.5 [53.6, 60.3]	52.3 [45.2, 56.5]	<0.001
Alpha Diversity Measures						
Observed OTUs	60.3 (13.9)	61.3 (13.5)	59.4 (12.6)	61.2 (14.9)	59.1 (14.7)	0.589
Pielou’s Evenness	0.6 [0.5, 0.7]	0.7 [0.6, 0.7]	0.6 [0.5, 0.7]	0.6 [0.5, 0.7]	0.6 [0.5, 0.7]	0.019
Shannon Diversity	2.5 (0.7)	2.6 (0.6)	2.5 (0.7)	2.5 (0.6)	2.3 (0.7)	0.037

Sample descriptive statistics are provided for *n* = 193 men from the Tobago Cohort Study who had complete data on computed tomography scans and fecal microbiota. Characteristics are presented as mean (standard deviation), median (interquartile range), or *n* (%). Linear trend *p*-values across quartiles of visceral adipose tissue were calculated using linear regression (for normally distributed continuous variables), Jonckheere–Terpstra test (for non-normally distributed continuous variables), Cochrane–Armitage trend test (for binary variables), or Cochran–Mantel–Haenszel test (for multi-level categorical variables). Abbreviations: OTU = operational taxonomic unit; HU = Hounsfield unit.

**Table 2 microorganisms-12-00812-t002:** Unadjusted and multivariable-adjusted standardized differences (95% CI) in BMI or fat measures according to a one standard deviation higher alpha diversity metric among men from the Tobago Health Study.

BMI or Fat Measure	Model	Observed OTUs	Pielou’s Evenness	Shannon Diversity Index
BMI	1	−0.06 (−0.21, 0.08)	**−0.19 (−0.33, −0.04)**	**−0.17 (−0.31, −0.03)**
	2	−0.02 (−0.16, 0.12)	**−0.16 (−0.30, −0.02)**	−0.14 (−0.27, 0.00)
VAT	1	−0.04 (−0.18, 0.11)	**−0.16 (−0.30, −0.02)**	−0.14 (−0.28, 0.00)
	2	−0.06 (−0.21, 0.09)	**−0.17 (−0.31, −0.04)**	**−0.15 (−0.29, −0.02)**
Pericardial fat	1	−0.01 (−0.16, 0.13)	−0.11 (−0.25, 0.03)	−0.10 (−0.24, 0.05)
	2	−0.03 (−0.18, 0.11)	−0.12 (−0.27, 0.02)	−0.11 (−0.25, 0.03)
Paraspinous IMAT	1	0.04 (−0.11, 0.18)	−0.04 (−0.19, 0.10)	−0.03 (−0.17, 0.12)
	2	−0.03 (−0.17, 0.12)	−0.09 (−0.23, 0.05)	−0.08 (−0.22, 0.06)
Psoas IMAT	1	−0.01 (−0.15, 0.14)	−0.07 (−0.22, 0.07)	−0.06 (−0.20, 0.08)
	2	−0.03 (−0.17, 0.11)	−0.08 (−0.22, 0.06)	−0.07 (−0.21, 0.07)
Thigh IMAT	1	−0.04 (−0.18, 0.10)	−0.09 (−0.23, 0.05)	−0.08 (−0.22, 0.06)
	2	−0.04 (−0.19, 0.10)	−0.09 (−0.24, 0.05)	−0.08 (−0.23, 0.06)
Paraspinous muscle attenuation	1	−0.06 (−0.20, 0.08)	−0.03 (−0.17, 0.11)	−0.04 (−0.18, 0.10)
	2	0.04 (−0.09, 0.16)	0.02 (−0.11, 0.15)	0.02 (−0.10, 0.15)
Psoas muscle attenuation	1	0.01 (−0.14, 0.15)	0.00 (−0.15, 0.14)	0.00 (−0.14, 0.14)
	2	0.07 (−0.08, 0.21)	0.03 (−0.11, 0.16)	0.04 (−0.10, 0.17)
Thigh muscle attenuation	1	−0.05 (−0.19, 0.09)	0.00 (−0.15, 0.14)	−0.01 (−0.16, 0.13)
	2	0.02 (−0.12, 0.17)	0.03 (−0.10, 0.17)	0.03 (−0.10, 0.16)
Liver attenuation	1	0.12 (−0.02, 0.26)	0.13 (−0.02, 0.27)	0.13 (−0.01, 0.27)
	2	0.09 (−0.05, 0.24)	0.10 (−0.04, 0.24)	0.10 (−0.04, 0.25)

A total of *n* = 193 men were included in the analyses. Fecal microbiota samples were first rarefied to a sequencing depth of 1500 reads. Estimates (95% CI) were based on multivariable linear regression models with BMI and fat measures as the dependent variables; models were analyzed separately by fat and by alpha diversity metric. Bold text indicates statistical significance at *p* < 0.05. Multivariable models included age (years), educational attainment (primary, secondary, or post-secondary), hours walked per week for exercise (hours), current smoking status (yes vs. no), drinking 4 or more alcoholic drinks per week (yes vs. no), the time difference between CT scans and fecal sample collection (years), and, in observed OTU models, unrarefied sequencing depth. Models were analyzed according to the following schema: M1: Unadjusted. The observed OTU model was additionally adjusted for sequencing depth. M2: M1 + age, education, hours walked per week for exercise, current smoking status, drinking 4+ alcoholic drinks/week, and the time difference between CT scan and fecal sample. Abbreviations: OTU = operational taxonomic unit; IMAT = intermuscular fat; CI = confidence interval; CT = computed tomography.

**Table 3 microorganisms-12-00812-t003:** The variance (adjusted R^2^) of the Bray-Curtis dissimilarity explained by BMI or fat measure among men from the Tobago Health Study.

BMI or Fat Measures	Adjusted R^2^	*p*-Value
BMI	**1.93%**	**0.0064**
VAT	**1.85%**	**0.0085**
Pericardial fat	1.14%	0.0519
Paraspinous IMAT	0.69%	0.2030
Psoas IMAT	0.75%	0.1649
Thigh IMAT	0.57%	0.2968
Paraspinous muscle attenuation	0.33%	0.7050
Psoas muscle attenuation	0.54%	0.3388
Thigh muscle attenuation	0.31%	0.7520
Liver attenuation	0.67%	0.2146

A total of *n* = 193 men were included in the analyses. Fecal microbiota samples were first rarefied to a sequencing depth of 1500 reads. Adjusted R^2^ and associated *p*-values are based on PERMANOVA models with BMI and fat measures as the independent variables of interest; models were analyzed separately by BMI or fat measure. Bold text indicates statistical significance at *p* < 0.05. Models adjusted for age (years), educational attainment (primary, secondary, or post-secondary), hours walked per week for exercise (hours), current smoking status (yes vs. no), drinking 4 or more alcoholic drinks per week (yes vs. no), and the time difference between CT scans and fecal sample collection (years). Abbreviations: PERMANOVA = permutational analysis of variance; IMAT = intermuscular fat; CT = computed tomography.

## Data Availability

The dataset analyzed in the current study may not be posted publicly in accordance with the original participant informed consent document. However, de-identified data may be requested from the parent Tobago Health study with an approved data use agreement through the University of Pittsburgh. The statistical code used to generate the results is available upon request from the corresponding author.

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
