# Peer review of "Associations of Fecal Microbiota with Ectopic Fat in African Caribbean Men"

_microorganisms, 2024, doi:10.3390/microorganisms12040812_

Round 1

Reviewer 1 Report

Comments and Suggestions for Authors

The manuscript by Curtis Tilves et al. investigated the associations of fecal microbiota with ectopic fat in African-Caribbean men. The study is interesting and the results are informative. I only have one suggestion. Table 1, table 2 and table 3 should be in a three-line format. This manuscript can be considered for publication. 

Reviewer 2 Report

Comments and Suggestions for Authors

In evaluating your paper 'Associations of Faecal Microbiota with Ectopic Fat in African-Caribbean Men', I found several aspects worthy of praise, such as the clarity of presentation and methodological approach. However, I have reservations about the paper's rationale and discussion.

Rationale: Although the association between the gut microbiome and visceral fat is acknowledged, the specific inclusion and focus on Afro-Caribbean men requires a more robust justification. It is suggested that the rationale be expanded to include additional pre-existing data or theories that support the importance of exploring this specific association in this population. It may be useful to consider comparative studies or theories linking dietary, genetic, or socioeconomic specificities to the composition of the gut microbiome and its impact on metabolic health in different ethnic groups.

Discussion: The discussion appears limited in its ability to link the findings to broader aspects of obesity, dietary habits, and the microbiome. I recommend supplementing the discussion with:

An in-depth reflection on the possible implications of your results for understanding the mechanisms underlying obesity and ectopic fat accumulation.

An analysis of how dietary habits might influence both the microbiome and fat accumulation, highlighting any data or theory linking specific dietary patterns to the composition of the gut microbiome in Afro-Caribbean men or similar populations.

Consideration of potential clinical implications of your findings, such as the possibility of microbiome-targeted interventions for the prevention or treatment of obesity and its complications in this specific population.

Finally, it would be helpful to discuss the limitations of your study in more detail, including considerations of participant self-selection and the influence of unmeasured confounding variables, such as specific lifestyle habits or pre-existing health conditions.

In summary, while your study adds important knowledge to the field, a broader discussion and better integration of the results into the existing context could increase its value and implication for future research and clinical practice. Translated with www.DeepL.com/Translator (free version)

Round 2

Reviewer 2 Report

Comments and Suggestions for Authors

The authors have modified the paper.

there are several shortcomings that need to be fixed before possible pubblication:

Broaden the comparison with previous studies: In the context of the introduction and discussion, a more detailed comparison with the results of previous studies, especially those conducted on populations of different ethnicities, would be useful. This would help to situate the new results in the existing scientific landscape, better highlighting the novelty and importance of the study's contribution.

More detail on the methods of statistical analysis: In the section on statistical methods, further details on the analysis procedures could be provided, including a more in-depth explanation of methodological choices and correction techniques for multiple testing, to increase the transparency and replicability of the study.

Discussion of clinical and future research implications: In the discussion, it would be useful to elaborate on the clinical implications of the results obtained and to provide more specific suggestions for future research. This could include discussions on how the results could influence clinical interventions or possible future studies to explore the mechanisms underlying the observed associations.

Examination of limitations: Although the paper acknowledges some limitations, a more detailed discussion of how these might have influenced the results and how they might be addressed in future studies could strengthen the robustness of the study. In particular, it would be useful to further explore the impact of selection. 

Considerations on population diversity: Given the importance of ethnic composition in microbiome research, the paper could benefit from a more in-depth discussion of the heterogeneity of the study population and the implications of these variations for the interpretation of results. A reflection on how genetic and cultural diversity may influence the results could enrich the discussion.

Comments on the Quality of English Language

English is fine
